# Improved Double Deep Q-Network Algorithm Applied to Multi-Dimensional Environment Path Planning of Hexapod Robots

**DOI:** 10.3390/s24072061

**Published:** 2024-03-23

**Authors:** Liuhongxu Chen, Qibiao Wang, Chao Deng, Bo Xie, Xianguo Tuo, Gang Jiang

**Affiliations:** 1School of Computer Science and Engineering, Sichuan University of Science and Engineering, Zigong 643000, China; 2School of Physics and Electronic Engineering, Sichuan University of Science and Engineering, Zigong 643000, China; 3School of Mechanical and Electrical Engineering, Chengdu University of Technology, Chengdu 610059, China

**Keywords:** hexapod robot, pathfinding, DDQN algorithm

## Abstract

Detecting transportation pipeline leakage points within chemical plants is difficult due to complex pathways, multi-dimensional survey points, and highly dynamic scenarios. However, hexapod robots’ maneuverability and adaptability make it an ideal candidate for conducting surveys across different planes. The path-planning problem of hexapod robots in multi-dimensional environments is a significant challenge, especially when identifying suitable transition points and planning shorter paths to reach survey points while traversing multi-level environments. This study proposes a Particle Swarm Optimization (PSO)-guided Double Deep Q-Network (DDQN) approach, namely, the PSO-guided DDQN (PG-DDQN) algorithm, for solving this problem. The proposed algorithm incorporates the PSO algorithm to supplant the traditional random selection strategy, and the data obtained from this guided approach are subsequently employed to train the DDQN neural network. The multi-dimensional random environment is abstracted into localized maps comprising current and next level planes. Comparative experiments were performed with PG-DDQN, standard DQN, and standard DDQN to evaluate the algorithm’s performance by using multiple randomly generated localized maps. After testing each iteration, each algorithm obtained the total reward values and completion times. The results demonstrate that PG-DDQN exhibited faster convergence under an equivalent iteration count. Compared with standard DQN and standard DDQN, reductions in path-planning time of at least 33.94% and 42.60%, respectively, were observed, significantly improving the robot’s mobility. Finally, the PG-DDQN algorithm was integrated with sensors onto a hexapod robot, and validation was performed through Gazebo simulations and Experiment. The results show that controlling hexapod robots by applying PG-DDQN provides valuable insights for path planning to reach transportation pipeline leakage points within chemical plants.

## 1. Introduction

Regular inspections and the detection of chemical pipeline leaks in chemical plants are paramount to safeguarding the environment from harmful chemical contamination, maintaining public health and safety, and reducing resource wastage and operational costs [1]. Conventionally, human personnel always conduct regular inspections. However, relying on human personnel for this task can be risky, as it increases their workload and exposes them to potential physical harm in the event of sudden pipeline leaks [2]. Hence, substituting human personnel with robots for this task is safer and more efficient, as it effectively prevents such occurrences. At present, some scholars have used robots to detect cracks. For example, the study [3] proposes the use of lidar and depth cameras to detect 3D cracks and build a robotic 3D crack detection system with an accuracy of <0.1 mm. However, choosing a suitable robot and an appropriate algorithm for path planning to inspect points within chemical plants is a fundamental prerequisite to address the challenge of chemical pipeline leak detection.

Hexapod robots have a distinct advantage over wheeled or tracked robots, as they offer a higher degree of freedom and greater reliance on discrete footholds for locomotion. Moreover, these robots demonstrate greater adaptability in complex environments than wheeled or tracked robots [4], making them ideal for patrolling chemical plants to detect pipeline leaks. However, the intricate layout of pipelines and the multi-dimensional nature of chemical plant spaces can make this inspection challenging, as hexapod robots might need to locate inspection points that are not confined to a single plane. It is imperative to accurately enable the hexapod robot to find the crossing point in the shortest crossing time and plan the shortest path across the plane to reach the inspection points within a multi-dimensional environment. However, this process is a pressing problem that requires resolution.

Previous studies have proposed various path-planning algorithms, falling into three categories [5,6,7]: population-based evolutionary algorithms, graph search planning algorithms, and artificial intelligence algorithms. Population-based evolutionary algorithms like the genetic algorithm (GA) and Ant Colony Optimization (ACO) are commonly used. Luo et al. [8] proposed the method of constructing unevenly distributed pheromones to solve the problem of slow convergence and low search efficiency. The effectiveness of the proposed method was verified by constructing a simulation grid map. This method pays more attention to the starting point and end point in path optimization problems. Another study [9], to improve the exploration capability of the genetic algorithm, incorporated Bezier curves into the algorithm to enhance the diversity of solutions generated by it. The study verified the feasibility of the algorithm through a small scene with 20 obstacles and 5 special maps.

Population-based evolutionary algorithms are commonly employed for optimizing paths between start and end points. However, in the multi-dimensional environment of chemical plants, there may exist multiple transition points on a given plane. It becomes essential to select these points based on their respective reward weights. Consequently, such algorithms may lack the environmental feedback characteristics required for formulating effective actions.

The A* algorithm is a well-known graph search planning algorithm. Zuo et al. [10] proposed a method aimed at enhancing a robot’s adaptability to unfamiliar maps by integrating least squares strategy iteration and the A* algorithm. This approach was validated on a simulation map characterized by multiple structures and extensive obstacle intervals, demonstrating the efficacy of the proposed technique. Li et al. [11] proposed a novel approach that incorporates a bidirectional alternating search strategy to enhance the A* algorithm. This method aims to solve the problems of extended computation time and excessive turning angles inherent in the A* algorithm by minimizing redundant nodes. Its effectiveness was validated on simulated maps with large structural obstacles as well as experimental maps. 

However, graph-based planning algorithms rely on known maps for planning, and the structure of the graph significantly impacts the performance of these algorithms. Given the dense network of pipelines within chemical plants, paths generated by such algorithms may not be suitable for hexapod robot locomotion, potentially resulting in collisions with the pipelines. Therefore, there is a need for path-planning algorithms capable of mapping to the actions of hexapod robots to facilitate their control.

The Q-learning algorithm and Deep Q-Network (DQN) algorithm are two well-known techniques in the field of artificial intelligence. Zou et al. [12] integrated the Q-learning algorithm with the Dynamic Window approach and the Mayfly Optimization algorithm to improve the algorithm’s global search capability and adaptability to complex environments. H. Quan et al. [13] enhanced the experience replay and sampling mechanisms within the DQN algorithm to augment its adaptability in dynamic environments. These algorithms can learn and adapt to intricate environments without prior knowledge and exhibit self-learning characteristics. As a result, they are well suited for six-legged robots to undertake path planning in multi-dimensional and complex environments.

Recently, some researchers have employed reinforcement learning algorithms for path planning in hexapod robots. X. Wang et al. [14] utilized the Soft-max function to enhance the DQN algorithm, thereby improving its adaptability for generating actions for hexapod robots. Additionally, L. Wang et al. [15] applied the Soft Actor-Critic algorithm for path planning in hexapod robots in outdoor environments, enhancing their robustness in unstructured environments.

One study [16] revealed that a reinforcement learning algorithm, DDQN, has demonstrated superior performance in decision-making problems with vast state spaces and challenging multi-dimensional environments that are notoriously difficult to model. DDQN uses two neural networks to overcome the overestimation problem, which greatly enhances its effectiveness. As a result, DDQN has been applied in numerous domains, including robot control and path planning, encompassing tasks such as legged robot control [17], unmanned aerial vehicle logistics and delivery [18], and path planning in unknown environments [19].

This study proposes an improved DDQN algorithm integrated with a hexapod robot for path planning for leak detection points in chemical plant pipelines. We used the global search capability and parallel computing ability of the PSO algorithm to overcome the limitations of traditional path-planning algorithms and replace the conventional random search strategy in the DDQN algorithm, thereby increasing DDQN’s ability to acquire environmental data quickly. In addition, we replaced the traditional BP neural network architecture with a four-input layer convolutional neural network structure to construct DDQN, which reduces the number of learning parameters. We abstracted the multi-dimensional environment into multiple grid-based planes to perform the preprocessing of the map data. These multiple grid-based planes consist of current and next layers as groups, dividing the overall path planning into multiple local planning to achieve map data preprocessing. We evaluated the effectiveness of the proposed method by employing subjective and objective evaluation metrics and making comparisons with traditional approaches. Then, we verified the feasibility of the proposed method through simulations and experiments.

## 2. PG-DDQN

This section delves into the utilization of the Particle Swarm Optimization (PSO)-guided Double Deep Q-Network (PG-DDQN) algorithm for multi-dimensional path planning with hexapod robots. Section 2.1 provides a comprehensive overview of a hexapod robot’s motion planning. Section 2.2 presents an in-depth explanation of the PG-DDQN algorithm.

### 2.1. Hexapod Robot’s Motion Planning

This study focuses on the fundamental tripod gait. In Section 3, the simulation and validation will primarily revolve around this gait pattern. The tripod gait is where three robot legs are consistently grounded while the remaining three legs are in an elevated position. The tripod gait pattern involves three distinct phases crucial to the robot’s motion.

Initial phase: At motion onset, the robot establishes stability by supporting itself on three legs, forming a triangular base in contact with the ground. The other three legs are in a swing phase, preparing to make ground contact for the subsequent step.

Transition phase: During this phase, the robot maneuvers the swing legs toward their target positions to establish contact with the ground. Three legs from the previous support base remain grounded, forming a new triangular support base. This phase involves motion control and action planning to guarantee seamless transitions and appropriate step lengths.

Final phase: In the last phase, the previously supporting legs are raised and transitioned into the swing phase, preparing for the next movement. Again, the other three legs simultaneously create a stable triangular support base, maintaining the robot’s stability. Hexapod robots achieve continuous and uninterrupted motion by executing these three phases recurrently.

The hexapod robot movement consists of forward and backward motions, lateral translations, and turning maneuvers. The illustration in Figure 1 shows movement details of this type of robot, with the supporting legs moving in the direction of the robot’s center of mass (COM) and the swing legs being lifted and moving in the opposite direction. The robot’s continuous motion is maintained by alternating the roles of the supporting and swinging legs.

### 2.2. Improved DDQN Algorithm

The PG-DDQN algorithm is presented as follows. The environment is processed, and the state space, action space, reward structure, and convolutional neural network are constructed. The PSO algorithm replaces the traditional random search strategy to speed up the accumulation of compelling data in the experience pool.

#### 2.2.1. DDQN Algorithm

The DDQN algorithm is a variant of the DQN algorithm [20] that leverages the practice of experience replay to enhance overall stability and performance, alleviating any potential correlations within the training data [21]. The parameters of the actual action value Q-network are trained in real-world scenarios. The target action value Q-network weights are periodically updated to reduce the overestimation’s influence on the training outcomes. The target value formulation for DDQN is as follows:(1)$$Q_{\mathrm{tagret}}=r+\gamma Q(s^{\prime },{\mathrm{argmax}}_{a}Q(s^{\prime },a|w)|w^{\prime })$$

The presented equation incorporates several essential components: γ, representing the discount factor; w, denoting the network structure weights in the Q-local neural network; $w^{\prime }$, corresponding to the network structure weights in the Q-target neural network; $s^{\prime }$, symbolizing the environmental state perceived at the next time step; a, signifying the selected action; and r, representing the reward value. Figure 2 depicts the schematic illustration of the DDQN model. PG-DDQN is improved on the basis of the DDQN algorithm.

#### 2.2.2. State Space

The robot must comprehensively understand its surroundings to navigate its environment effectively. A hexapod robot gathers extensive global information, such as scene layout, comprising dynamic obstacles, static obstacles, navigable waypoints, self-position, target location, and operational status. Four matrices are formulated, based on the extracted current coordinate information from the environment, to represent the prevailing environmental state: the Path Matrix (PM), Obstacle Matrix (OM), Location Matrix (LM), and Target Matrix (TM). The environment is discretized into a grid, with each matrix corresponding to one or more grid cells, where each position in the matrices adopts either of two values: 0, signifying false, and 1, signifying true. Pertinent information is captured in a grid map when extracting a planar section from the multi-dimensional environment, producing four distinct matrices. Figure 3 showcases how the hexapod robot’s motion model uses these matrices to observe the current state. Then, the state vector of the hexapod, s, can be expressed by concatenating the aforementioned four vectors as follows:(2)s=[PM,OM,LM,TM]

#### 2.2.3. Action Space

In the subsequent experimental section, we demonstrate a hexapod robot’s specific movements in a grid-based environment, where its motion is broken down into four categories, i.e., up, down, left, and right, facilitating effective path planning. Each action is associated with a numerical code, devoid of inherent meaning but serving as an identifier due to the numerical nature of neural network outputs (Figure 4).

#### 2.2.4. Reward Structure

Rewards are critical to providing feedback regarding the hexapod robot’s efficacy and performance according to the algorithm, as they help evaluate the actions’ worth and effectiveness.
(3)$$r\left(s,a\right)=r_{1}\left(s,a\right)+r_{2}\left(s,a\right)+r_{3}\left(s,a\right)+r_{4}\left(s,a\right)+r_{5}\left(s,a\right)+r_{pso}\left(s,a\right)$$

The instantaneous reward value comprises five parts: $r_{1}$ indicates that the hexapod moves outside the given map, $r_{2}$ indicates the penalty for collision with obstacles, $r_{3}$ indicates the reward value for selecting the target point, and $r_{4}$ indicates a one-time reward for reaching the end point. $r_{pso}$ means that PSO accumulates rewards/penalties.

We define $r_{1}$ as follows:(4)r1=−100 n=1−120 n>10    n=0
where n represents the number of times the hexapod robot touches the boundary during its movement. For robots that leave the map multiple times, a larger penalty value must be given to avoid errors in state evaluation, so the penalty value is here set to 120. $r_{2}$ gives the penalty for collisions with obstacles:(5)$$r_{2}=\left\{\begin{array}{l} -20\;\;if({\left(x,y\right)}_{h\mathrm{ex}apod},dob)\\ -50\;\;if({\left(x,y\right)}_{h\mathrm{ex}apod},sob)\\ 0\;\;\;other \end{array}\right.$$
where $\left({\left(x,y\right)}_{h\mathrm{ex}apod},dob\right)$ is the indicator function showing whether the designated hexapod position collides with a dynamic obstacle and $\left({\left(x,y\right)}_{h\mathrm{ex}apod},sob\right)$ means that the hexapod collides with a static obstacle.

This study introduces two types of crossing points, i.e., slope-type and trapezoid-type points, with varying reward values. $r_{3}$ represents the crossing point reward value: The trapezoid-type transition points have reward values of 25, 30, 35, and 40, while the slope-type transition points have reward values of 30, 35, and 40. $r_{4}$ is associated with a reward value of 40. $r_{5}$ is defined as follows:(6)r5−25  ifdst−2>d(st)0 else
where $d(s_{t})$ denotes the distance between the current state and the target or end point, while $d\left(s_{t-2}\right)$ represents the distance between the state from two-time steps ago and the transition or end point.
(7)rpso=−10  if(ρ>5&dt−1<dt)0 other
where ρ represents the number of particles in this state, and $d_{t}$ represents the distance between the current position and the target point.

#### 2.2.5. Neural Networks

Our input data comprise two-dimensional plane data with a layout size of 40 × 40, representing the current state coordinating and transforming into four observation matrices. The neural network architecture is divided into three main components based on these input data. Figure 5 illustrates the architecture of the neural network.

The first step involves feeding the matrix into the neural network’s convolutional layer, which comprises 16 convolutional kernels, each with a size of 5 × 5 and a stride of 1 (stride = 1). Then, the input data are padded with an additional border of two units with zero values (padding = 2). Through the convolution operation, we obtain 16 feature matrices of size 40 × 40, passing through a subsequent max-pooling layer with a pooling window size of 2 × 2 and a stride of 2 × 2, resulting in a 16 × 20 × 20 feature matrix.

In the second step, the 16 × 20 × 20 feature matrix serves as input data, fed into another convolutional layer. This layer comprises 32 convolutional kernels, each with a size of 5 × 5 with the same stride and padding settings as in the first step. The convolution operation produces 32 feature matrices of size 20 × 20, passing through another max-pooling layer with a pooling window size of 2 × 2 and a stride of 2 × 2, resulting in a 32 × 10 × 10 feature matrix.

In the third step, the data obtained from the first two components are processed through a linear fitting, with 3200 input nodes passing through three fully connected layers, ultimately yielding 1 × 4 one-dimensional output data. Neural network parameters will be adjusted in simulation and real-life environments, but the neural network structure will not change.

#### 2.2.6. PG-DDQN Algorithm

In the DDQN algorithm, random exploration is vital for enriching the decisions made by the algorithm, as the neural network consistently makes the same decisions in response to repetitive observations. Random exploration also facilitates diversity and allows robots to make different decisions, thereby increasing the exploration capacity. However, this approach can be inefficient for path planning in hexapod robots, especially in multi-dimensional environments mapped to grid-based maps.

As depicted in Figure 6, Panel A displays the hexapod robot approaching a transition point with static obstacles above and below. In this scenario, only the action “Right” is the correct choice, as the actions “Up” and “Down” are incorrect. The action “Left” leads the hexapod robot away from the transition point. In this case, random exploration has a 50% chance of making an incorrect decision, a 25% chance of making a suboptimal decision, and only a 25% chance of making the correct decision.

As shown in Panel B in Figure 6, the hexapod robot must enter the transition point. Random exploration in this scenario has a 25% chance of making an illegal decision, a 50% chance of making a suboptimal decision, and only a 25% chance of making the correct decision. This scenario also has a 25% chance of terminating the motion model in Panel B.

We have introduced the PSO algorithm to replace random exploration to enhance the efficiency of exploration and make the latter more purposeful. 

The PSO algorithm proceeds as follows:

Step 1: Data preparation and initialization. First, the obstacle data are constructed using accumulated experience in the knowledge pool. Next, the positions and velocities of the particles in the swarm are initialized with each particle i, and its position vector $X_{i}$ and velocity vector $V_{i}$ are randomly initialized. Additionally, each particle’s individual best position $P_{i}$ is set initially to its current position $X_{i}$. The global optimal position (G) is initialized; the selection of G requires a combination of angle and distance screening with the final target point. First, it is necessary to find the angle and distance between each particle and the final target point. The formula is as follows:(8)$$\left\{\begin{array}{c} d=\sqrt{{\left(y_{i}+y_{g}\right)}^{2}+{\left(x_{i}+x_{g}\right)}^{2}}\\ ag=\arctan(\frac{\left|y_{i}-y_{g}\right|}{\left|x_{i}+x_{g}\right|}) \end{array}\right.$$
where $\left(x_{i},y_{i}\right)$ represents the initialized particle coordinates in the frame-selected map data, $\left(x_{g},y_{g}\right)$ represents the global map target coordinates, d represents the distance, and ag represents the angle between the particle and the target.
(9)$$\left\{\begin{array}{l} G=rand(C_{rand}<K\&(ag\in (-\frac{\pi }{3},\frac{\pi }{3})))\begin{array}{cc} if & K\neq 0 \end{array}\\ G=\underset{d}{\min}(x,y)\begin{array}{c} \;\;\;\;\;\;\;\;\;else \end{array} \end{array}\right.$$
where $C_{rand}$ represents the collection of particle coordinates contained in the random map boundary particle coordinates as the center of the circle. K represents the exploration rate, which changes with iteration time.

Step 2: Updating of particle velocity and position. For each particle i, the velocity vector is updated using the following equation:(10)$$V_{i}\left(t+1\right)=\omega \times V_{i}\left(t\right)+c_{1}\times rand()\times \left(P_{i}-X_{i}\right)+c_{2}\times rand()\times \left(G-X_{i}\right)$$
where $V_{i}\left(t\right)$ represents the velocity of particle i at time t, ω is the inertia weight, $c_{1}$ and $c_{2}$ are the learning factors, and rand() is a function that generates random numbers. Subsequently, the position vector is updated as follows:(11)$$X_{i}\left(t+1\right)=X_{i}\left(t\right)+V_{i}\left(t+1\right)$$

The fitness of the new position, $X_{i}\left(t+1\right)$, is evaluated, and the individual’s best position, $P_{i}$, is updated if the fitness of $X_{i}\left(t+1\right)$ is superior to that of $P_{i}$.

Step 4: Iteration. The process of updating particle velocities and positions (Step 2) and updating the global best position (Step 3) is repeated until the stopping condition is met, which may involve reaching the maximum number of iterations or finding a satisfactory solution.

The action selection strategy is as follows:(12)$$\pi (s_{t})=\left\{\begin{array}{c} \underset{a}{\arg\max}Q(s_{t+1},a;w)\\ \tilde{A} \end{array}\right.\begin{array}{c} \;if\;\;P\geq K\\ \;\;else \end{array}$$
where P is a randomly generated number between 0 and 1 with 8 decimal points in each round and $\tilde{A}$ represents the choice of the actions by PSO.
(13)$$y_{t}=r_{t+1}+\gamma Q^{-}\left(s_{t+1},\arg\max_{a^{\prime }}Q(s_{t+1},a^{\prime };w);w^{\prime }\right)$$
where the value of γ is between 0 and 1, $a^{\prime }$ represents the next action selected by using the Q-local neural network, and $y_{t}$ represents the actual value of the optimization target network.
(14)$$\left\{\begin{array}{l} L(\theta )=\frac{1}{2}{\left(Q(s_{t},a_{t};w)-y_{t}\right)}^{2}\\ \theta \leftarrow \theta -\alpha \nabla_{\theta }L\left(\theta \right) \end{array}\right.$$

Finally, when a certain number is reached, the Q-target neural network is updated. Figure 7 depicts the PG-DDQN training structure on the abstract plane grid motion model, indicating that the training process consists of several steps.

## 3. Experiment and Discussion

This section presents an overview of the experiments conducted and the conclusions drawn from them. We performed all experiments on a computer system equipped with an Intel(R) Core(TM) i9-13900HX CPU operating at 2.20 GHz alongside an NVIDIA GeForce RTX 4060 Laptop GPU. An abstraction approach was employed to address multi-dimensional environments, decomposed into a composition of multiple current and next layers. Each transition from the current layer plane to the next was regarded as an instance of localized path planning, and this technique was used in conjunction with other instances to create comprehensive path planning for multi-dimensional environments. We adopted a localized path-planning approach involving transitions from the current to the next layer plane to evaluate the algorithm’s performance. This strategy effectively applied localized planning across various multi-dimensional environments, since it could be replicated based on the dimension of the environment. By evaluating the algorithm’s performance within this localized context, we could assess its applicability effectively.

To facilitate the experimentation, we skillfully constructed two-dimensional planes using Python’s TKINTER library for the current and next layer planes. Implementing the PG-DDQN algorithm, which is the core of our research, was meticulously undertaken in Python, leveraging the PYTORCH framework. We validated simulation results based on the simulations conducted on Ubuntu 20.04, utilizing Gazebo 11 as the designated platform.

### 3.1. Algorithm Comparison

We employed a random obstacle generation method to generate the maps, yielding three sets of current and subsequent layer planes to compose them, and comprehensive information is presented in Figure 8 and Table 1. The left image shows the map of the current plane, while the right image depicts that of the next plane. Each layer plane has dimensions of 40 × 40, incorporating four dynamic obstacles. The exploration rates for each map are specified in Table 1. The total number of blocks, encompassing dynamic obstacles, static obstacles, transition points, and target points, denotes the abstract size of the maps.

We employed two distinct performance metrics to measure the algorithm’s overall performance.

(1)Overall reward value

The reward value for each training set is calculated based on the reward acquisition rules, with the maximum achievable score for the current layer and the next layer’s final destination point being valued at 40 points. Each plane is equipped with transition points. The current layer plane contains seven transition points; the next layer represents the destination points. The hexapod robot can obtain the total reward value by completing the task without penalty. This process is calculated as follows:(15)$$Total\;reward=reward_{current\;layer}+reward_{next\;layer}$$

(2)Time to complete the path

The timing of the robot’s journey is initiated at its starting position, and the total time taken to traverse the entire planned path is measured. We apply a threshold range to the total reward value to mitigate the prolonged evaluation of erroneous paths. The path completion time is zero when the reward value surpasses this threshold. The precise calculation formula is as follows:(16)Ttotal=Tend−Tstart   Step≤Stepmax 0         Else

In PG-DDQN training, “training” refers to updating the local neural network to achieve convergence. Table 2 displays the actual statistics of training steps. During training, the algorithm compares the total reward value after each completed epoch to a constraint. If the total reward falls below −5000, the current epoch test is stopped, and a reward value of −5000 is recorded based on this constraint.

These hyperparameters are based according to the references provided by [22]. The learning rate is initially set to 0.0025, and when every 10% of the training is completed, the learning rate is reduced to 95% of the original. The discount factor determines the extent to which future rewards influence current decisions. If the discount factor approaches 1, the agent tends to consider long-term rewards, whereas if the discount factor approaches 0, the agent focuses more on immediate rewards. The PG-DDQN parameter settings are shown in Table 3.

The performance of the PG-DDQN algorithm on three random maps is shown in the Figure 9.

In Figure 9, the left image represents the map of total reward, while the right image depicts the map of total movement time. The x-axis represents the number of training epochs, while the y-axis represents the cumulative reward obtained from testing after each epoch. The figure shows that the entire neural network requires approximately 1850 epochs to achieve convergence for each map.

The results from Figure 10 indicate that the enhanced PG-DDQN algorithm outperforms the standard DQN and DDQN algorithms regarding overall total reward across all three random maps. Additionally, the standard DQN and DDQN algorithms exhibit weaker convergence when abstracting three-dimensional environments into two-dimensional maps than the improved PG-DDQN algorithm. By comparing map 1 and map 2 in Figure 9, it can be seen that higher exploration rates and increased PSO guidance lead to more stable training amplitudes during the training phase and enhance more excellent stability during the oscillation period. Table 4 presents the differences in computational time among the algorithms. Additionally, comparing completion times reveals that the improved algorithm achieves shorter path-planning times than the other two methods.

Figure 11 shows the algorithm’s success rate in each random map without touching obstacles and completing path planning. It can be seen that the growth trend of the success rate of the improved algorithm is greater than the standard DDQN and DQN algorithms, and the success rate of the improved algorithm is greater than the standard DDQN and DQN algorithms.

### 3.2. Simulation Verification

The simulation validation was conducted using the Gazebo simulation platform, setting up the simulation environment as depicted in Figure 12. The dual-layer simulation environment represents the current and subsequent layers for simulation validation. Figure 13 shows the simulation model, with the left side representing the mobile obstacle model, implemented using a wheeled robot as a reference for the dynamic obstacles. The right side of Figure 13 shows the hexapod robot model.

In the Gazebo environment, the PG-DDQN algorithm was integrated with the ROS system to validate performance after training, where path planning is depicted in Figure 14. In this validation process, the algorithm can handle step-type points for plane traversal. The black box represents the hexapod robot, while the red lines illustrate the paths planned by the algorithm. The orange squares in the current layer indicate the entrances to incline-type point 1, and those in the subsequent layer, the exits from incline-type point 1. Similarly, the blue squares in the current layer mark the entrances to incline-type point 2, while in the following layer, they represent the exits from incline-type point 2. Furthermore, the green squares in the current layer indicate the entrances to step-type points, while in the next layer, they indicate the exits from the step-type points.

Figure 15 and Figure 16 illustrate the planned trajectory of the hexapod robot during traversal in the current and following layers, respectively.

During traversal in the current layer, the hexapod robot can avoid obstacles encountered during movement, as demonstrated in Figure 17. In Step 1, the hexapod robot detects a moving obstacle and promptly halts its movement. In Step 2, it assesses the trajectory of the moving obstacle to determine the best course of action. Once the path is evaluated, the hexapod robot resumes its movement, adhering to the planned path in Step 3. The robot makes necessary adjustments to its body path from Steps 4 to 7 to effectively navigate the obstacle. The path-planning process within the moving obstacle’s range is successfully executed from Steps 9 to 12.

In this section, we have developed the following conclusions by comparing the algorithm and validating simulations: The improved PG-DDQN algorithm performs better than the other two reinforcement learning algorithms regarding convergence stability and achieving optimal rewards while path planning in abstract local environments in multi-dimensional scenarios. Moreover, the application of the PG-DDQN algorithm, in conjunction with sensor integration, for path planning in hexapod robots demonstrates the effectiveness of this improved approach in efficiently navigating through multi-dimensional terrains within the simulated environment.

### 3.3. Experimental Verification

The experimental verification used the laboratory indoor environment to simulate the chemical plant scene. The roads in the scene are complex; the site environment is shown in Figure 18.

Due to site restrictions, the site area was divided in order to build a multi-dimensional environment, and different goals that needed to be reached were set in different stages to achieve the setting of a multi-dimensional environment, as shown in Figure 19.

In the current layer, the hexapod robot successfully avoided interference target points during movement, as shown in Figure 20. 

Then, the hexapod then successfully avoided dynamic obstacles. The movement process is shown in Figure 21.

Finally, the target point position of the current layer was reached, as shown in Figure 22.

At this time, the hexapod robot entered the next layer and moved to the target point position of the next layer, as shown in Figure 23.

In this section, we present the following conclusions drawn through actual testing: As it can be seen from the verification diagram above, the robot did not hit stationary obstacles when moving. When encountering interference points, the robot’s route can be changed to avoid them, and dynamic obstacles can also be avoided. The feasibility of this improved approach in effectively navigating multi-dimensional terrain is demonstrated.

## 4. Conclusions

This study addresses the challenges encountered in path planning for detecting chemical plant pipeline leaks, presenting a hexapod robot path-planning method based on PG-DDQN. The proposed method exhibits the following characteristics:

(1) To address problems in multi-dimensional environments, we employ the preprocessing of the multi-dimensional environment to divide the data of path planning. We abstract the multi-dimensional environment into multiple grid-based planes, aiming to optimize data processing and enhance planning efficiency.

(2) The PSO algorithm is implemented as a guiding strategy to supplant the random selection strategy in action selection, augmenting the accumulation of meaningful data within the experience pool and enhancing the algorithm’s convergence rate.

(3) Introducing the four-input layer convolutional neural network also accelerates the processing speed of high-dimensional information.

The proposed method effectively addresses the challenges hexapod robots face in path planning for detecting chemical plant pipeline leaks, such as dynamic environmental scenes, complex road structures, and multi-dimensional survey points. Compared with the standard DQN and DDQN algorithms, this approach demonstrates faster convergence and reduces path-planning time by at least 33.94%. The feasibility of the algorithm was verified through Gazebo simulations and experiments. This path-planning method has important practical value for pipeline leak detection in chemical plants. There is room for improvement in the algorithm’s completion rate in environments with more obstacles. Follow-up work will continue to improve the algorithm’s adaptability to larger maps and combine it with the hexapod robot’s gait to achieve path planning in rugged terrain; this study provides new ideas for the future multi-robot coordinated processing of pipeline repair and chemical material processing.

## Figures and Tables

**Figure 1 sensors-24-02061-f001:**
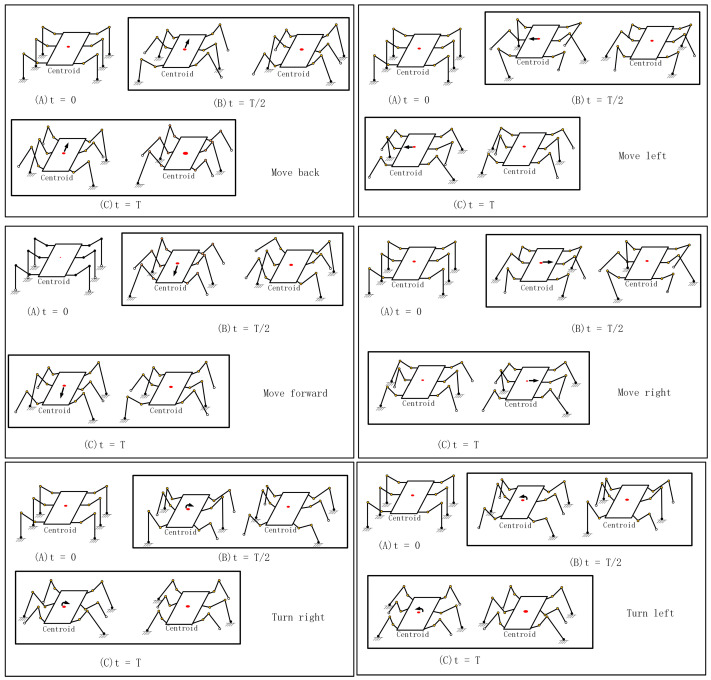
Hexapod robot moving forward and backward. The red dot indicates the center of mass of the hexapod robot, and the arrow indicates the direction of movement of the center of mass.

**Figure 2 sensors-24-02061-f002:**
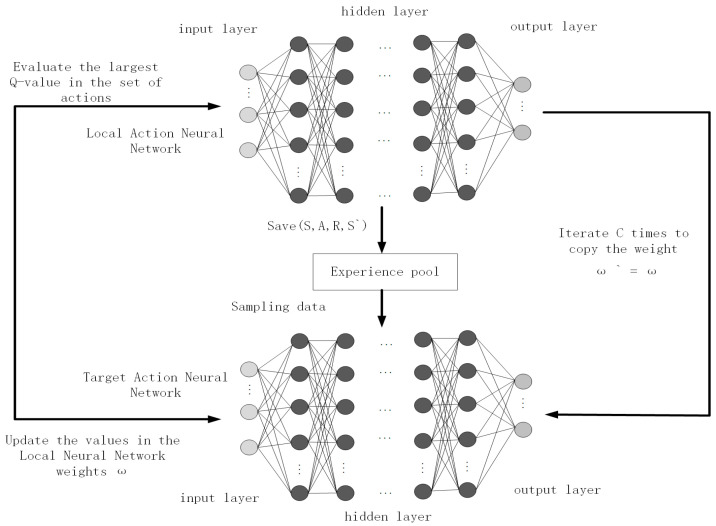
DDQN algorithm structure diagram.

**Figure 3 sensors-24-02061-f003:**
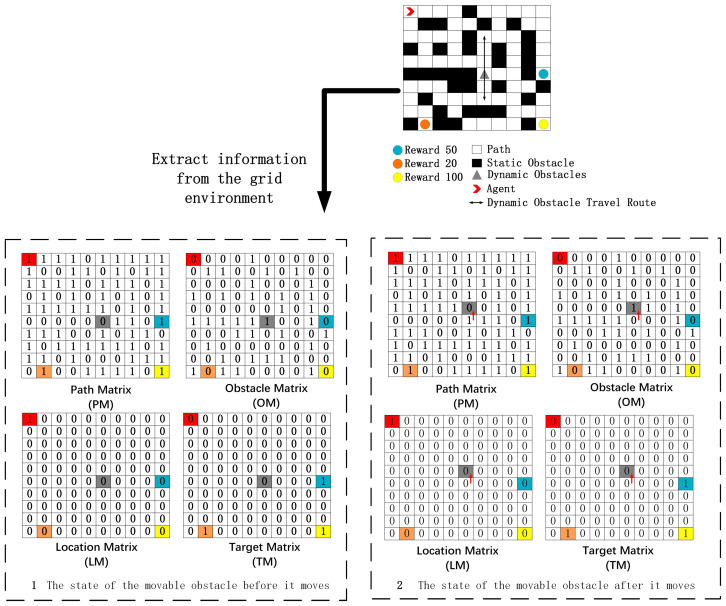
An example of observation of the current state of a hexapod robot kinematic model. In the numbers, red represents "Agent", orange represents "reward 20", yellow represents "reward 100", blue represents "reward 50", gray represents "moving obstacle", and the red arrow represents the moving direction of the moving obstacle.

**Figure 4 sensors-24-02061-f004:**
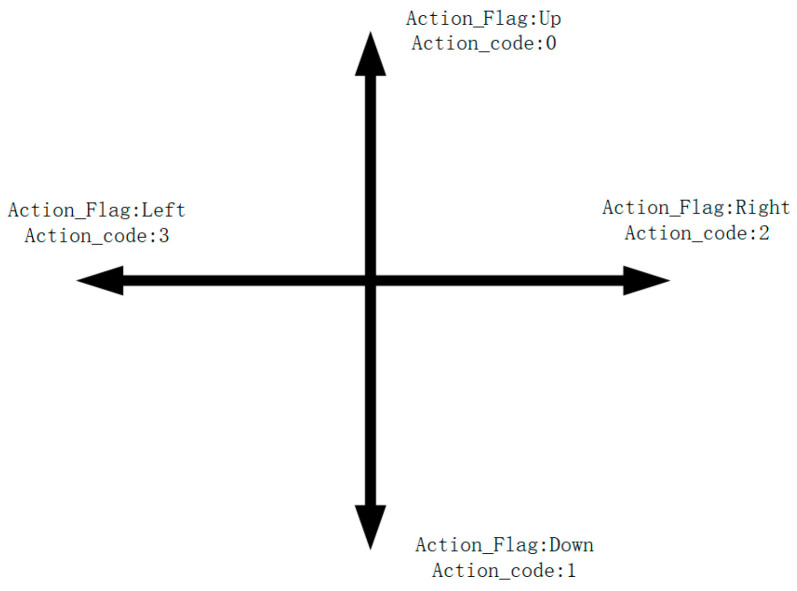
Neural network output actions.

**Figure 5 sensors-24-02061-f005:**
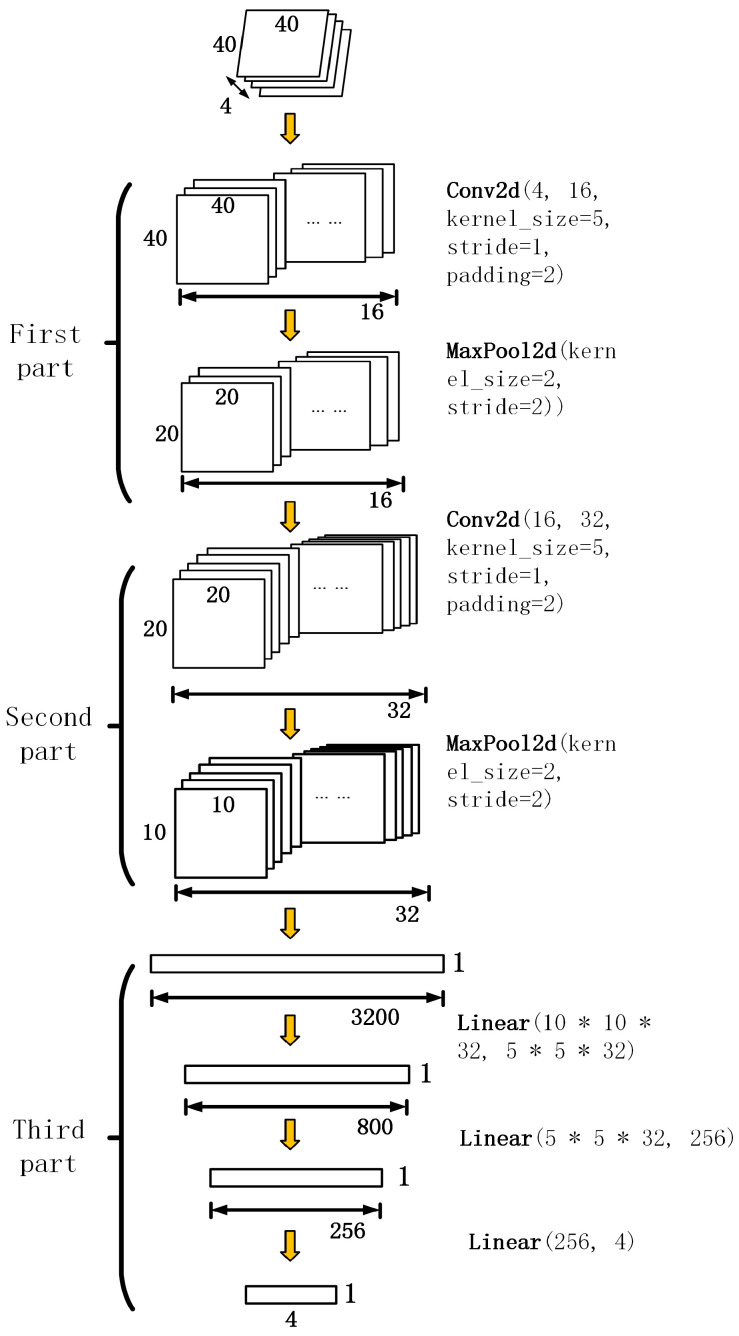
Neural network architecture.

**Figure 6 sensors-24-02061-f006:**
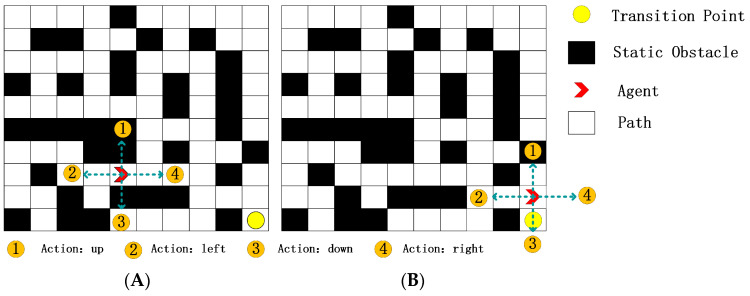
The picture on the left is Panel (**A**), and the picture on the right is Panel (**B**).

**Figure 7 sensors-24-02061-f007:**
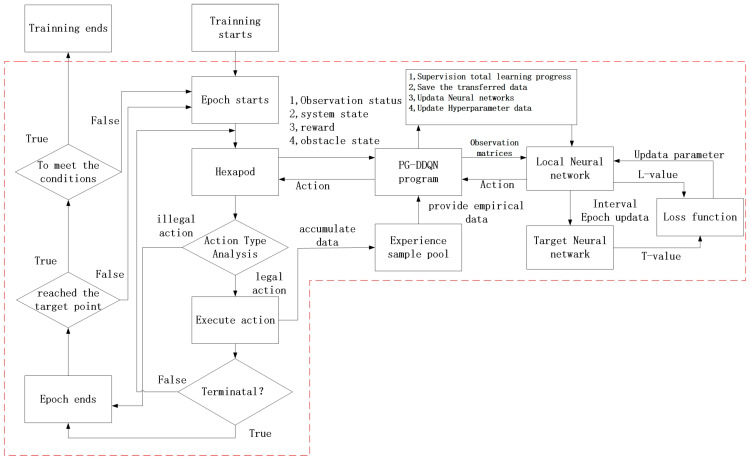
Training structure of PG-DDQN on abstract maps.

**Figure 8 sensors-24-02061-f008:**
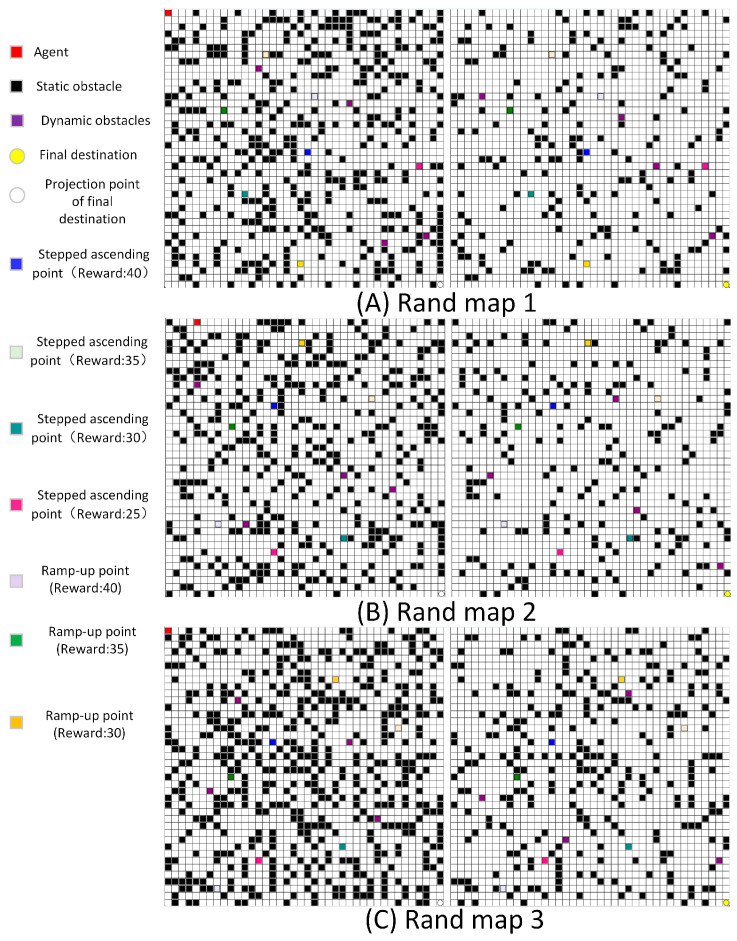
Abstract maps.

**Figure 9 sensors-24-02061-f009:**
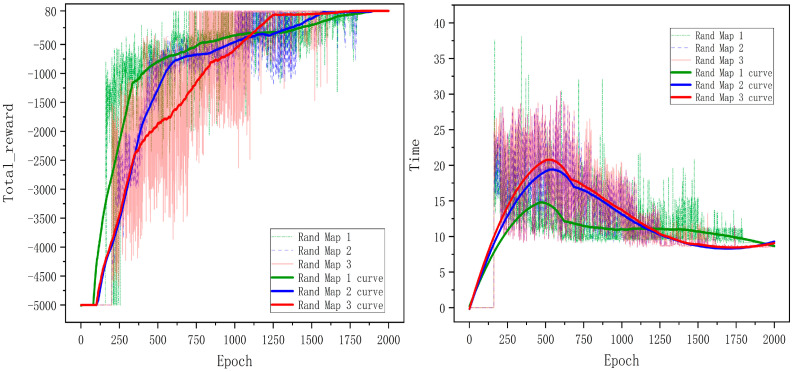
The performance of the PG-DDQN algorithm.

**Figure 10 sensors-24-02061-f010:**
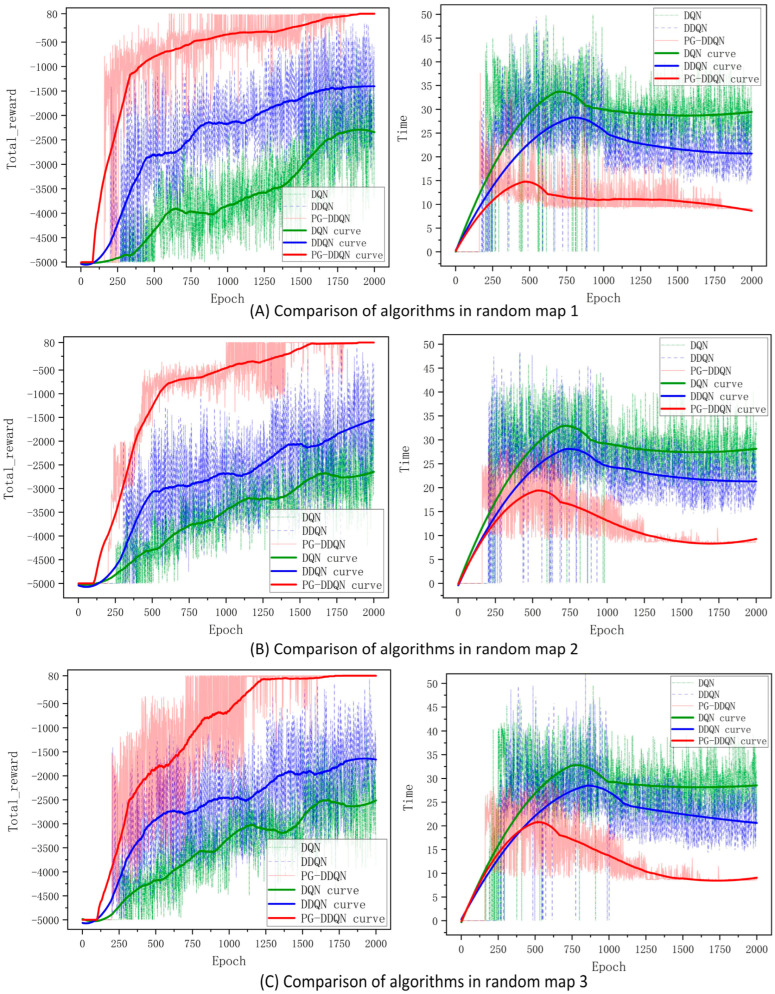
Comparison of reward value and time of different algorithms on random maps.

**Figure 11 sensors-24-02061-f011:**
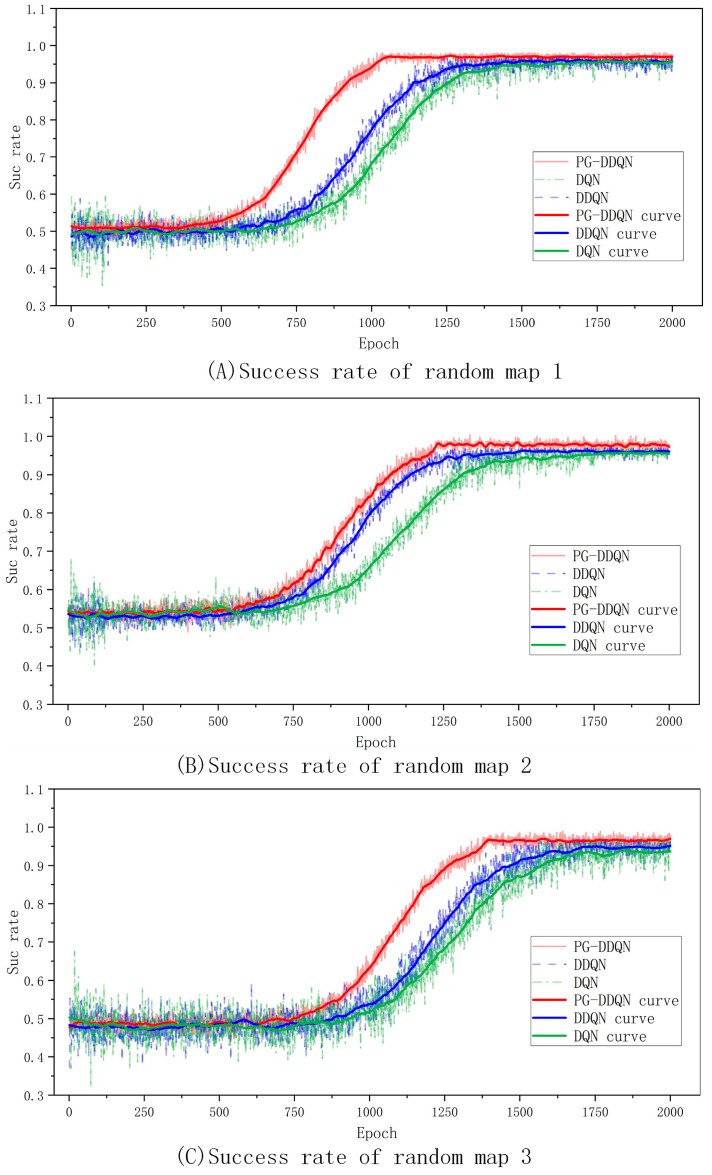
Comparison of success rates of different algorithms on random maps.

**Figure 12 sensors-24-02061-f012:**
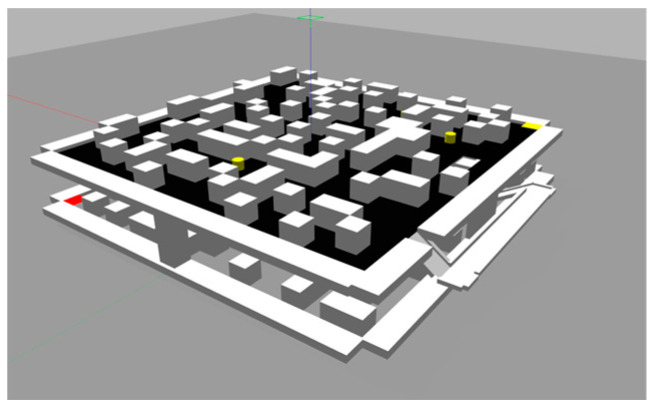
Simulation environment diagram.

**Figure 13 sensors-24-02061-f013:**
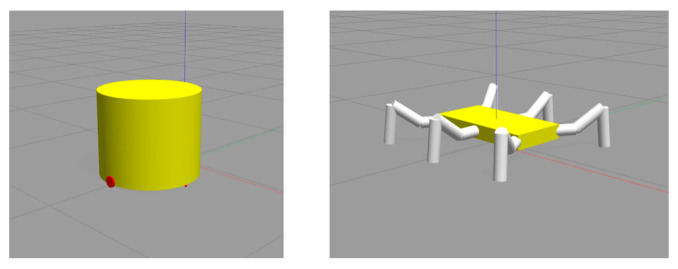
Model diagram in simulation verification.

**Figure 14 sensors-24-02061-f014:**
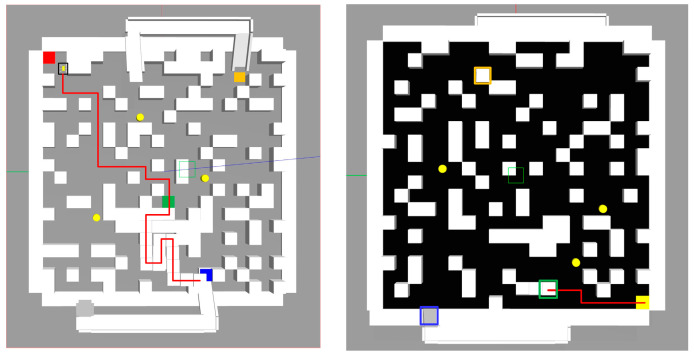
Hexapod robot path-planning diagram. The red square represents the “starting position”, the yellow square represents the “end position”, the red line represents the “planned path”, and the yellow circle represents the “dynamic obstacles”.

**Figure 15 sensors-24-02061-f015:**
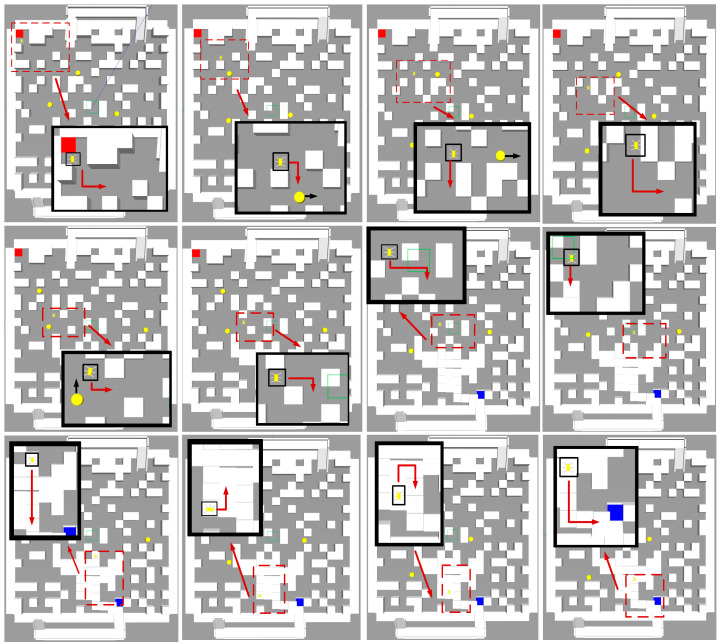
Schematic diagram of the current floor plan in path planning. The red line frame represents the area that needs to be enlarged, the black line frame represents the enlarged image of the area selected by the red line frame, the red square represents the starting point, the blue square represents the current layer of the incline-type point, the yellow circle represents the dynamic obstacles, and the red arrow represents the moving direction of the hexapod robot, the black arrow indicates the moving direction of the dynamic obstacles.

**Figure 16 sensors-24-02061-f016:**
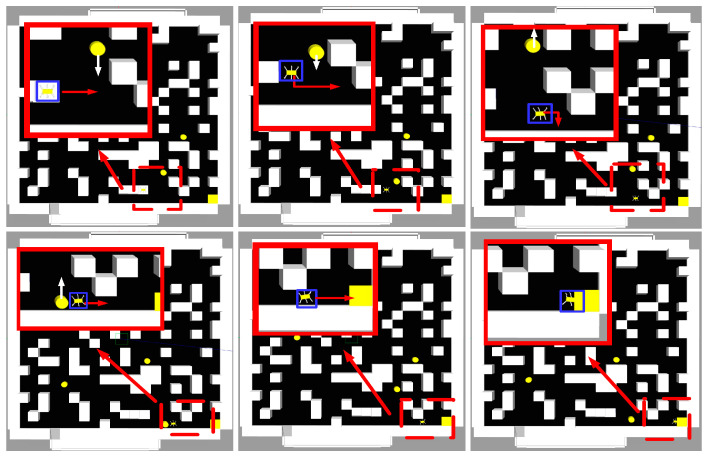
Schematic diagram of the next floor plan in path planning. The red dotted line box represents the area that needs to be enlarged, the red box represents the enlarged image of the selected area of the red dotted line box, the blue line box represents the hexapod robot, the yellow square represents the end point, the yellow circle represents the dynamic obstacles, and the red arrow represents the movement of the hexapod robot direction, the white arrow represents the moving direction of the dynamic obstacles.

**Figure 17 sensors-24-02061-f017:**
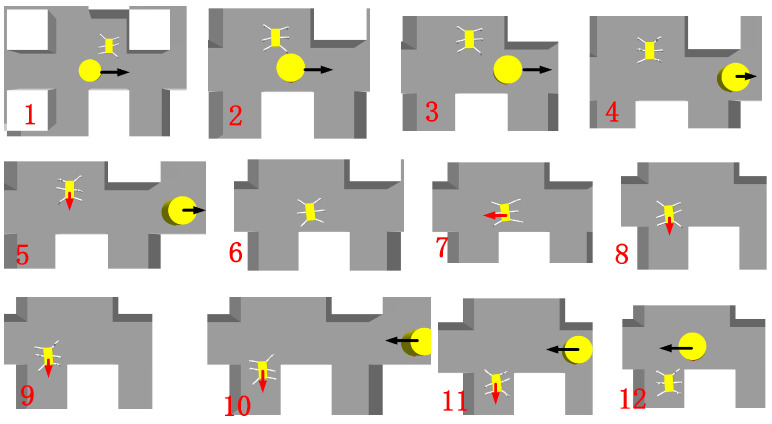
Schematic diagram of hexapod robot avoiding moving obstacles. The red arrow indicates the moving direction of the hexapod robot, and the black arrow indicates the moving direction of the dynamic obstacles.

**Figure 18 sensors-24-02061-f018:**
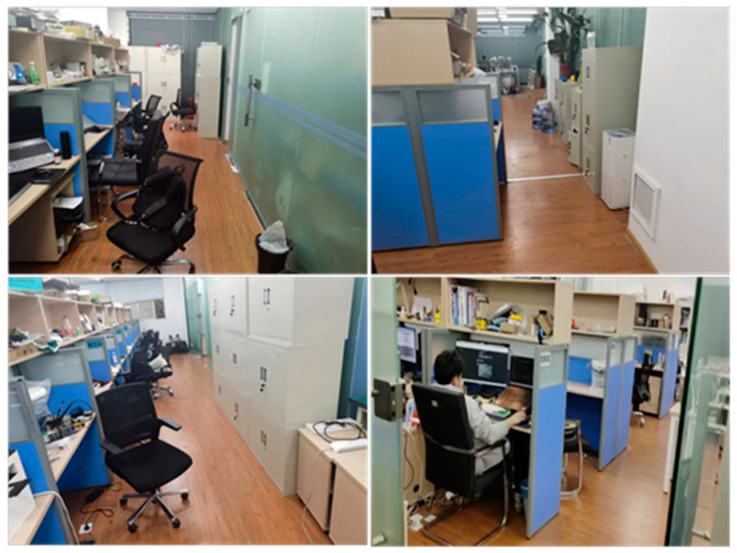
Test environment.

**Figure 19 sensors-24-02061-f019:**
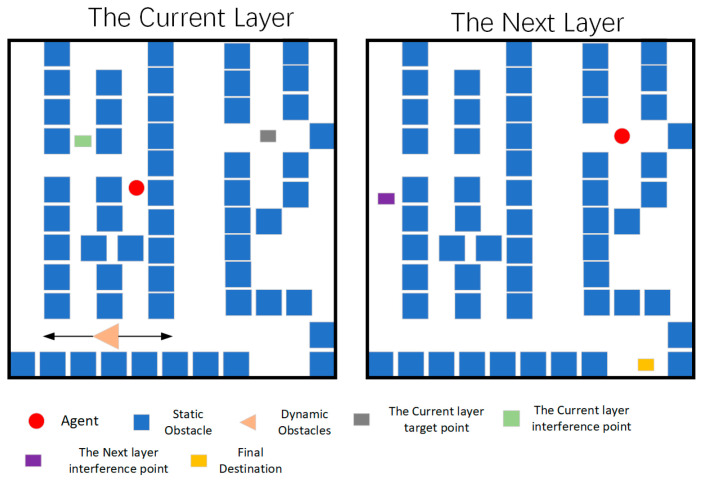
Grid map of test environment. The black arrow indicates the moving range of the movable obstacle.

**Figure 20 sensors-24-02061-f020:**
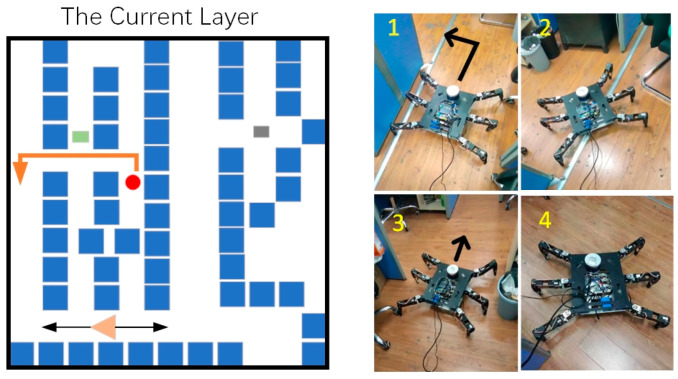
The first stage in the current layer. The orange line represents the movement path of the hexapod robot at this stage, and (**1**–**4**) show the specific movement details of the hexapod robot.

**Figure 21 sensors-24-02061-f021:**
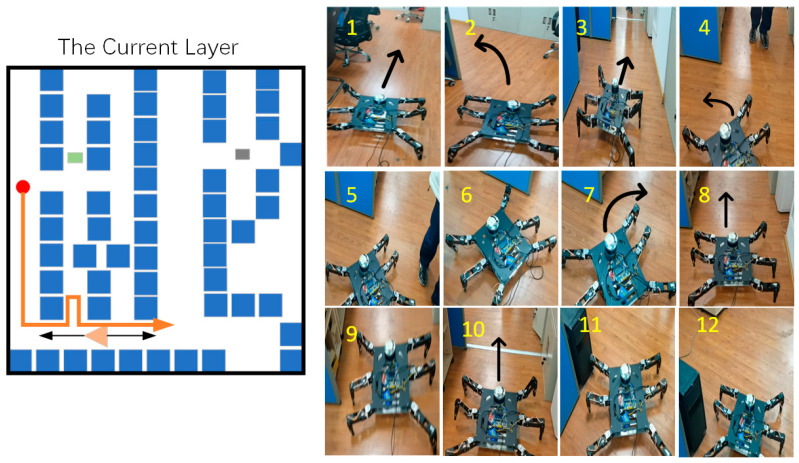
The second stage in the current layer. The orange line represents the movement path of the hexapod robot at this stage, and (**1**–**12**) show the specific movement details of the hexapod robot.

**Figure 22 sensors-24-02061-f022:**
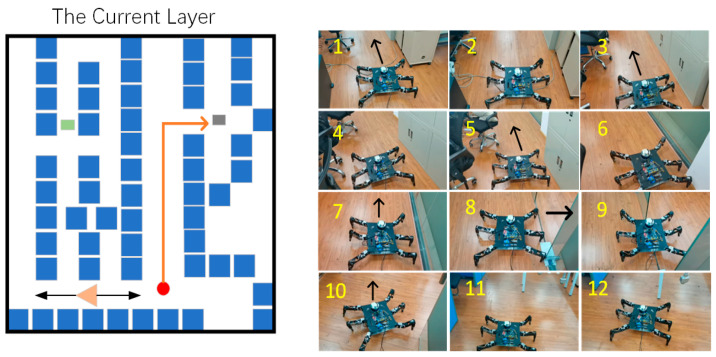
The third stage in the current layer. The orange line represents the movement path of the hexapod robot at this stage, and (**1**–**12**) show the specific movement details of the hexapod robot.

**Figure 23 sensors-24-02061-f023:**
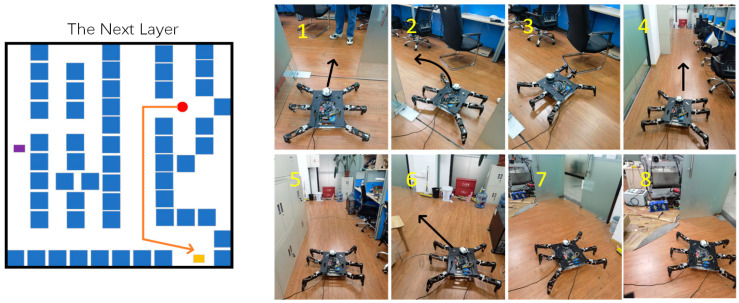
Movement process in the next layer. The orange line represents the movement path of the hexapod robot at this stage, and (**1**–**8**) show the specific movement details of the hexapod robot.

**Table 1 sensors-24-02061-t001:** Layout details of the three abstract maps.

Layout	Number of Static Obstacles (Current Layer)	Number of Static Obstacles (Next Layer)	Exploring Rate
Random map 1	376	193	0.6
Random map 2	376	193	0.8
Random map 3	498	272	0.8

**Table 2 sensors-24-02061-t002:** Actual training times for three random maps.

Layout	Actual Number of Steps in Training
Random map 1	15,000,000
Random map 2	20,000,000
Random map 3	30,000,000

**Table 3 sensors-24-02061-t003:** Hyperparameters for the proposed PG-DDQN algorithm.

Parameter	Value
Learning rate (α)	0.0025
Discount factor (λ)	0.99
Batch size	3000
Time step	600,000

**Table 4 sensors-24-02061-t004:** Performance of different algorithms on random maps.

Map	Algorithm	Spent Time	Success Rate	Total Step	Time Reduction
Rand1	PG-DDQN	8.9593	97.65%	96	37.96%/45.49%
	DDQN	14.4432	96.85%	144	
	DQN	16.4361	95.99%	165	
Rand2	PG-DDQN	8.6003	97.36%	93	33.94%/43.82%
	DDQN	13.0194	97.05%	136	
	DQN	15.3101	96.21%	153	
Rand3	PG-DDQN	8.5154	96.89%	99	34.60%/42.60%
	DDQN	13.0209	96.34%	152	
	DQN	14.8359	96.01%	176	

## Data Availability

No data were used for the research described in the article.

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
