# Peer review of "Improved Double Deep Q-Network Algorithm Applied to Multi-Dimensional Environment Path Planning of Hexapod Robots"

_sensors, 2024, doi:10.3390/s24072061_

Round 1
Reviewer 1 Report
Comments and Suggestions for Authors
Based on the document content, here are some suggestions for comments and remarks:
General Feedback:
- Well-written paper with a clearly defined problem statement, proposed methodology, experimental results, and conclusions. The paper is logically structured and easy to follow.
- The abstract provides a nice concise summary of the key elements of the paper.
- Good literature review covering the relevant existing work on path planning algorithms.
- The explanations and illustrations of the hexapod robot motion planning and gait patterns are clear.
- The details provided on the state space formulation, action space, reward structure and neural network architecture are appropriate.
- The comparative experimental results demonstrate the improved convergence and time savings of the proposed PG-DDQN algorithm over DQN and DDQN.
- The simulation and experimental verification provide validation of the algorithm's effectiveness.
Suggested Improvements:
- Consider expanding the related work section to include some recent papers on hexapod robot path planning and navigation using deep reinforcement learning.
- Provide some more insight into how the hyperparameter values like learning rate, discount factor etc. were tuned.
- Include some sample trained neural network weights or architecture diagrams to give more specifics.
- Expand the conclusion to summarize the key contributions and mention potential future work like multi-robot coordination.
- Carefully proofread the paper to fix any minor grammatical/spelling errors.
Please let me know if you would like any clarification or have additional suggestions for my comments!
Author Response
Dear Reviewer,
On behalf of all the contributing authors, I would like to express our sincere appreciation for your time and the professional comments concerning our article entitled “Improved Double Deep Q-Network algorithm applied to the multi-dimensional environment path planning of hexapod robot” (Manuscript No. sensors-2905505). These detailed and constructive comments help to improve the academic rigor of our article. According to your suggestions and comments, we have carefully made extensive modifications and supplemented extra data to make our results convincing. In this revised version, changes to our manuscript were highlighted within the document in red-colored text. Point-by-point responses are listed below this letter.
Point 1: Consider expanding the related work section to include some recent papers on hexapod robot path planning and navigation using deep reinforcement learning.
Response 1: Thank you for underlining this deficiency, we sincerely appreciate for your professional comments and realized the inadequacy of this part. Based on your suggestions, we have added research on reinforcement learning used for path planning of hexapod robots. Please refer to page 3 of the revised manuscript, lines 105-110.
How the paper is modified: For your review, we have attached the revised results: “Currently, some researchers are employing reinforcement learning algorithms for path planning in hexapod robots. Wang X et al. [14] utilized the Soft-max function to enhance the DQN algorithm, thereby improving its adaptability in generating actions for hexapod robots. Additionally, Wang L et al. [15] applied the Soft Actor-Critic algo-rithm for path planning of hexapod robots in outdoor environments, enhancing their robustness in unstructured environments.” (Page 3, lines 105-110.)
Point 2: Provide some more insight into how the hyperparameter values like learning rate, discount factor etc. were tuned.
Response 2: Thank you for pointing this out. We did mention it in Section 3.1, but it may not be clear. Hence, we have rewritten the relevant sentences on page 13, lines 393-398.
How the paper is modified: For your review, we have attached the revised results. This part is now modified as follows: “These hyperparameters are based according to the references provided by [22]. The learning rate is initially set to 0.0025, and when every 10% of the training is completed, the learning rate is reduced to 95% of the original. Discount factor determines the extent to which future rewards influence current decisions. If the discount factor approaches 1, the agent tends to consider long-term rewards; whereas if the discount factor approaches 0, the agent focuses more on immediate rewards.” (Page 13, lines 393-398.)
Point 3: Include some sample trained neural network weights or architecture diagrams to give more specifics.
Response 3: Thanks for pointing this out. In this paper, we have added a structural diagram of the neural network for the convenience of readers. Therefore, we added the relevant content of the neural network structure diagram on page 7, lines 250-252.
How the paper is modified: For your review, we have attached the revised results. The
details are as follows: “Figure 5 illustrates the architecture of the neural network.
Figure 5. Neural Network Architecture.” (Page 7, lines 250-252.)
Point 4: Expand the conclusion to summarize the key contributions and mention potential future work like multi-robot coordination.
Response 4: Thank you for pointing this out and for your professional review work on our article. We did not provide a point-by-point summary of the key contributions and did not clearly explain the possible future work, so we made modifications in the summary. The detailed text is located on lines 515-523 on page 20 and 532-535 on page 21 of the revised manuscript.
How the paper is modified: For your review, we have attached the revised results. This part is now modified as follows: “1, Addressing problems in multidimensional environments, we employ the pre-processing of the multi-dimensional environment to divide the data of path planning. We abstract the multi-dimensional environment into multiple grid-based plane, aiming to optimize data processing and enhance planning efficiency.
2, Employing the PSO algorithm as a guiding strategy is implemented to supplant the random selection strategy in action selection, augmenting the accumulation of meaningful data within the experience pool and enhancing the algorithm's convergence rate.
3, Introducing the four-input layer convolutional neural network also accelerates the processing speed of high-dimensional information.
Follow-up work will continue to improve the algorithm's adaptability to other larger maps and combine it with the hexapod robot's gait to achieve path planning in rugged terrain, and it provides new ideas for future multi-robot coordinated processing of pipeline repair and chemical material processing.” (Page 20, lines 515-523, and Page 20, lines 532-535.)
Point 5: Carefully proofread the paper to fix any minor grammatical/spelling errors.
Response 5: Thank you for your comments. We carefully proofread the paper and corrected grammatical/spelling errors. No grammatical/spelling errors were found.
At last, thank you very much for your time involved in reviewing the manuscript. We hope that the revisions we have made address the concerns raised and improve the quality of the manuscript. If there are any other modifications we could make, we would like very much to modify them and we really appreciate your help.
Thank you again for your time and attention. We look forward to hearing from you soon.
Sincerely,
Qibiao Wang

Reviewer 2 Report
Comments and Suggestions for Authors
In the domain of pipeline leak detection in chemical plants, particularly in addressing the complexities of path planning in multi-dimensional environments, an efficient path planning method holds crucial significance. In this context, the researchers have devised a novel high-performance path planning algorithm termed PG-DDQN, which integrates Particle Swarm Optimization and Double Deep Q-Network to improve the navigation success rate of robots in intricate settings. Leveraging the particle swarm optimization algorithm as a bootstrap strategy, the algorithm accelerates convergence speed and processes high-dimensional information by incorporating a four-input layer convolutional neural network. Upon experimental validation, the algorithm demonstrates efficient path planning on a hexapod robot platform, significantly enhancing the efficiency and success rate of path planning compared to the standard DQN and DDQN algorithms.
1. When discussing the limitations of existing path planning algorithms, the authors should provide additional real-world examples or data on the application of these algorithms in the context of chemical plant pipeline leak detection to further illustrate their shortcomings.
2. It is advisable for the authors to clarify whether the "boundary" in Equation 4 can be interpreted as a "static obstacle" to better differentiate between the two.
3. The paper should provide additional explanation and clarification regarding the penalty in Equation 4, addressing the reasonableness of setting the penalty at -120 regardless of the number of contacts with the boundary.
4. The authors are encouraged to offer a detailed analysis of the limitations and future improvements of the techniques studied to enhance the reader's understanding of the study's results and their potential applications.
5. Supplementing the discussion on the obstacle avoidance rate is recommended to evaluate the effectiveness of the algorithm in dealing with dynamic and static obstacles. The paper should provide data and analysis on the obstacle avoidance rate to address this aspect.
6. Figures 3, 5, 6, and 9 in the paper lack clear labeling; it is suggested that the authors redraw these figures to ensure that all labels and legends are legible for better reader comprehension.
7. The authors should discuss additional literature related to obstacle avoidance navigation to augment the comprehensiveness of the paper. For instance, references such as "Obstacle Avoidance Motion in Mobile Robotics; Journal of System Simulation" and "3D vision technologies for a self-developed structural external crack damage recognition robot; Automation in Construction" could be included.
Author Response
Dear Reviewer,
On behalf of all the contributing authors, I would like to express our sincere appreciation for your time and the professional comments concerning our article entitled “Improved Double Deep Q-Network algorithm applied to the multi-dimensional environment path planning of hexapod robot” (Manuscript No. sensors-2905505). These detailed and constructive comments help to improve the academic rigor of our article. According to your suggestions and comments, we have carefully made extensive modifications and supplemented extra data to make our results convincing. In this revised version, changes to our manuscript were highlighted within the document in red-colored text. Point-by-point responses are listed below this letter.
Point 1: When discussing the limitations of existing path planning algorithms, the authors should provide additional real-world examples or data on the application of these algorithms in the context of chemical plant pipeline leak detection to further illustrate their shortcomings.
Response 1: Thank you for underlining this deficiency. We sincerely appreciate the valuable comment and realized the inadequacy of this part. According to your suggestions, we have modified the discussion of other algorithms to explain their current application scenarios and to further explain the limitations of these algorithms. Please read page 2 of the revised manuscript, lines 65 to 94
How the paper is modified: For your review, we have attached the revised results. The
details are as follows: “Luo et al. [8] proposed a method of constructing unevenly distributed pheromone to solve the problem of slow convergence and low search efficiency. The effectiveness of the proposed method is verified by constructing a simulation grid map. This method pays more attention to the starting point and end point path optimization problems. Another study [9], to improve the exploration capability of the algorithm, incorporated Bezier curves into the genetic algorithm to enhance the diversity of solutions generated by the algorithm. This method verifies the feasibility of the algorithm through a small scene with 20 obstacles and 5 special maps.
Population-based evolutionary algorithms are commonly employed for optimizing paths between starting and ending points. However, in the multi-dimensional environment of chemical plants, there may exist multiple transition points within a given plane. It becomes essential to select these points based on their respective reward weights. Consequently, such algorithms may lack the environmental feedback characteristics required for formulating effective actions. Zuo et al. [10] proposed a method aimed at enhancing a robot's adaptability to unfamiliar maps by integrating the least squares strategy iteration with the A* algorithm. This approach was validated on a simulation map characterized by multiple structures and extensive obstacle intervals, demonstrating the efficacy of the proposed technique. Li et al. [11] proposed a novel approach that incorporates a bidirectional alternating search strategy to enhance the A* algorithm. This method aims to solve the problems of extended computation time and excessive turning angles inherent in the A* algorithm by minimizing redundant nodes. Its effectiveness has been validated on simulated maps with large structural obstacles as well as experimental maps.
However, graph-based planning algorithms rely on known maps for planning, and the structure of the graph significantly impacts the performance of these algorithms. Given the dense network of pipelines within chemical plants, paths generated by such algorithms may not be suitable for hexapod robot locomotion, potentially resulting in collisions with the pipelines. Therefore, there is a need for path-planning algorithms capable of mapping to the actions of hexapod robots to facilitate their control.” (Page 2, lines 65-94.)
Point 2: It is advisable for the authors to clarify whether the "boundary" in Equation 4 can be interpreted as a "static obstacle" to better differentiate between the two.
Response 2: Thank you for underlining this deficiency. We may not have expressed it clearly enough. By "boundary" we mainly mean that the robot walks outside the map, so this state needs to be penalized. We have re-described the role of reward value r1, see page 7 of the revised manuscript, lines 219-220.
How the paper is modified: For your review, we have attached the revised results. The
details are as follows: “r1 indicates that the hexapod moves outside the given map” (Page 7, lines 219-220.)
Point 3: The paper should provide additional explanation and clarification regarding the penalty in Equation 4, addressing the reasonableness of setting the penalty at -120 regardless of the number of contacts with the boundary.
Response 3: Thank you for underlining this deficiency. We may not have expressed it clearly enough. For the robot to leave the map multiple times, we need to give a larger penalty value to avoid misestimation of the state, so we set 120 as the reward value. We have modified the explanation of 120 in the manuscript, please see the revised manuscript on Page 7, lines 227-228.
How the paper is modified: For your review, we have attached the revised results. The
details are as follows: “For robots that leave the map multiple times, a larger penalty value must be given to avoid errors in state evaluation, so the penalty value is set here to 120.” (Page 7, lines 219-220.)
Point 4: The authors are encouraged to offer a detailed analysis of the limitations and future improvements of the techniques studied to enhance the reader's understanding of the study's results and their potential applications.
Response 4: Thank you for underlining this deficiency. According to your suggestions, we present a detailed analysis of the limitations and future improvements of the studied technology and describe our future work, see page 21, lines 529-535 of the revised manuscript.
How the paper is modified: For your review, we have attached the revised results. The
details are as follows: “This path planning method has important practical value for pipeline leak detection in chemical plants. There is room for improvement in the algorithm's completion rate in more obstacle environments. Follow-up work will continue to improve the algorithm's adaptability to other larger maps and combine it with the hexapod robot's gait to achieve path planning in rugged terrain, and it provides new ideas for future multi-robot coordinated processing of pipeline repair and chemical material processing.” (Page 21, lines 529-535.)
Point 5: Supplementing the discussion on the obstacle avoidance rate is recommended to evaluate the effectiveness of the algorithm in dealing with dynamic and static obstacles. The paper should provide data and analysis on the obstacle avoidance rate to address this aspect.
Response 5: Thank you for underlining this deficiency. According to your suggestion, we use the completion rate of path planning without collision with obstacles to indicate whether the hexapod robot collides with obstacles. The completion rate of each cycle is averaged to reflect the obstacle avoidance effect of the entire hexapod robot on a random map. A comparison chart of the three algorithms, as shown in Figure 11, can more clearly reflect the changes, and the completion rate is added in Table 4. Please refer to lines 422-430 on page 15 of the revised manuscript.
How the paper is modified: For your review, we have attached the revised results. The
details are as follows: “
Figure 11. The performance of the PG-DDQN algorithm.
Figure 11 shows the algorithm's success rate in each random map without touching obstacles and completing path planning. It can be seen that the growth trend of the success rate of the improved algorithm is greater than the standard DDQN and DQN algorithms, and the success rate of the improved algorithm is greater than the standard DDQN and DQN algorithms.
Table 4. Layout details of the three abstract maps.
Map |
Algorithm |
Spent time |
Success rate |
Total step |
Save time |
Rand1 |
PG-DDQN |
8.9593 |
97.65% |
96 |
37.96%/45.49% |
DDQN |
14.4432 |
96.85% |
144 |
||
DQN |
16.4361 |
95.99% |
165 |
||
Rand2 |
PG-DDQN |
8.6003 |
97.36% |
93 |
33.94%/43.82% |
DDQN |
13.0194 |
97.05% |
136 |
||
DQN |
15.3101 |
96.21% |
153 |
||
Rand3 |
PG-DDQN |
8.5154 |
96.89% |
99 |
34.60%/42.60% |
DDQN |
13.0209 |
96.34% |
152 |
||
DQN |
14.8359 |
96.01% |
176 |
” (Page 15, lines 422-430.)
Point 6: Figures 3, 5, 6, and 9 in the paper lack clear labeling; it is suggested that the authors redraw these figures to ensure that all labels and legends are legible for better reader comprehension.
Response 6: Thank you for underlining this deficiency. According to your suggestion, we have redrawn Figures 3, 5, 6 and 9 and enlarged all labels and legends for better understanding by readers.
Point 7: The authors should discuss additional literature related to obstacle avoidance navigation to augment the comprehensiveness of the paper. For instance, references such as "Obstacle Avoidance Motion in Mobile Robotics; Journal of System Simulation" and "3D vision technologies for a self-developed structural external crack damage recognition robot; Automation in Construction" could be included.
Response 7: Thank you for underlining this deficiency. According to your suggestions, we have added a summary of pipeline detection and current path planning algorithms, see lines 44-49 on page 1 and lines 61-63 on page 2 of the revised manuscript.
How the paper is modified: For your review, we have attached the revised results. The
details are as follows: “At present, some scholars have used robots to detect cracks. For example, literature [3] proposes the use of lidar and depth cameras to detect 3D cracks and build a robotic 3D crack detection system with a crack detection accuracy of <0.1mm. However, choosing a suitable robot and an appropriate algorithm for path planning to inspect points within chemical plants is a fundamental prerequisite to address the challenge of chemical pipeline leak detection. Previous studies have proposed various path-planning algorithms, falling into three categories [5-7]: population-based evolutionary algorithms, graph search planning algorithms, and artificial intelligence algorithms.” (Page1, lines 44-49, and Page2, lines 61-63.)
At last, thank you very much for your time involved in reviewing the manuscript. We hope that the revisions we have made address the concerns raised and improve the quality of the manuscript. If there are any other modifications we could make, we would like very much to modify them and we really appreciate your help.
Thank you again for your time and attention. We look forward to hearing from you soon.
Sincerely,
Qibiao Wang
